# Role of a RhoA/ROCK-Dependent Pathway on Renal Connexin43 Regulation in the Angiotensin II-Induced Renal Damage

**DOI:** 10.3390/ijms20184408

**Published:** 2019-09-07

**Authors:** Gonzalo I. Gómez, Victoria Velarde, Juan C. Sáez

**Affiliations:** 1Instituto de Ciencias Biomédicas, Facultad de Ciencias de la Salud, Universidad Autónoma de Chile, El Llano Subercaseaux #2801, Santiago 8910060, Chile; 2Departamento de Fisiología, Facultad de Ciencias Biológicas, Pontificia Universidad Católica de Chile, Alameda #340, Santiago 8331150, Chile; 3Instituto de Neurociencias, Centro Interdisciplinario de Neurociencias de Valparaíso, Universidad de Valparaíso, Valparaíso 2381850, Chile

**Keywords:** Hypertensive nephropathy, oxidative stress, fibrosis, inflammation, Cx43, Fasudil

## Abstract

In various models of chronic kidney disease, the amount and localization of Cx43 in the nephron is known to increase, but the intracellular pathways that regulate these changes have not been identified. Therefore, we proposed that: “In the model of renal damage induced by infusion of angiotensin II (AngII), a RhoA/ROCK-dependent pathway, is activated and regulates the abundance of renal Cx43”. In rats, we evaluated: 1) the time-point where the renal damage induced by AngII is no longer reversible; and 2) the involvement of a RhoA/ROCK-dependent pathway and its relationship with the amount of Cx43 in this irreversible stage. Systolic blood pressure (SBP) and renal function (urinary protein/urinary creatinine: Uprot/UCrea) were evaluated as systemic and organ outcomes, respectively. In kidney tissue, we also evaluated: 1) oxidative stress (amount of thiobarbituric acid reactive species), 2) inflammation (immunoperoxidase detection of the inflammatory markers ED-1 and IL-1β), 3) fibrosis (immune detection of type III collagen; Col III) and 4) activity of RhoA/ROCK (amount of phosphorylated MYPT1; p-MYPT1). The ratio Uprot/UCrea, SBP, oxidative stress, inflammation, amount of Cx43 and p-MYPT1 remained high 2 weeks after suspending AngII treatment in rats treated for 4 weeks with AngII. These responses were not observed in rats treated with AngII for less than 4 weeks, in which all measurements returned spontaneously close to the control values after suspending AngII treatment. Rats treated with AngII for 6 weeks and co-treated for the last 4 weeks with Fasudil, an inhibitor of ROCK, showed high SBP but did not present renal damage or increased amount of renal Cx43. Therefore, renal damage induced by AngII correlates with the activation of RhoA/ROCK and the increase in Cx43 amounts and can be prevented by inhibitors of this pathway.

## 1. Introduction

In the last years Chronic kidney disease (CKD) has been increasing steadily worldwide turning into a public health problem. In fact, about 10% of adults in developed countries suffer some degree of kidney damage [1]. Patients with this disease, usually develop a progressive kidney damage characterized by tubulointerstitial fibrosis and/or glomerular sclerosis, eventually leading to the last stage of this condition, end-stage renal disease [2]. In this process there is an increase in damaged nephrons which leads to a progressive reduction of glomerular filtration rate (GFR), and eventually to organ failure [3]. Different etiologies, including glomerulonephritis, hypertensive nephrosclerosis and diabetic nephropathy, can produce CKD. However, regardless of the initial cause, similar morphological characteristics are observed, such as glomerular sclerosis and tubular necrosis [4,5]. 

Hypertension is one of the most common complications that predisposes patients to other health problems and is the second cause of terminal CKD [6,7]. Hypertensive nephropathy starts in the glomeruli due to an increase in intra-glomerular pressure. These initial events activate and damage mesangial cells, epithelial cells and podocytes within the glomerulus. In turn, these cells produce vasoactive and pro-inflammatory mediators, which increase cell damage and favor fibrosis, reducing renal blood flow, permeability, and glomerular filtration [8]. The renin angiotensin system (RAS) is the prototype of a classic systemic endocrine network that acting on the kidneys and on the adrenal glands, regulate blood pressure, intravascular volume and electrolyte balance [9]. The vasoconstriction and increase in blood pressure mediated by angiotensin II (AngII) represent only part of the pleiotropic actions of this peptide. AngII stimulates aldosterone secretion, cell infiltration, proliferation and migration, thrombosis, superoxide ion production and other alterations in nephropathy [8]. In addition, high concentrations of AngII maintained for long periods of time, induce an inflammatory response characterized by the expression of pro-inflammatory cytokines such as interleukin-1β (IL-1β) and tumor necrosis factor-α (TNF-α) [10,11], infiltration of macrophages (ED-1 positive cells) [12], tubular overexpression of osteopontin (OPN) [12], and the expression of other pro-inflammatory cytokines, chemokines and adhesion molecules, which are ultimately associated with kidney damage induced by AngII [8,13]. Moreover, AngII increases the expression of nicotinamide adenine dinucleotide phosphate (NADPH) oxidase (NOX), one of the main enzymes in the generation of reactive oxygen species (ROS), contributing to the onset of oxidative stress (OS), independent of the action of pro-inflammatory cytokines [14,15]. OS has been proposed to be involved in several pathological states such as cardiovascular diseases, infections, cancer, diabetes, neurodegenerative disorders and also in the progression of kidney damage [16]. Intrarenal RAS is an important factor in the pathophysiology of hypertension and hypertensive nephropathy [17]. Therefore, hypertension induced by the administration of AngII is a widely used model in the study of kidney damage, because in addition to providing a clinically relevant model of systemic hypertension, it produces progressive damage in the nephrons, which ultimately generates a chronic lesion, that eventually leads to end stage renal disease [8,17,18].

Two receptors, AngII receptor type 1 and type 2 (AT1 and AT2 receptors), both coupled to a G-protein, mediate the actions of AngII. The AT1 receptor activates small G proteins, including Ras, Rac1, RhoA and the Rho kinase system (ROCK) [19], while the AT2 receptor inhibits RhoA [20]. The Rho family of small GTPases (Rho GTPase) is constituted by monomeric G proteins between 20 and 40 kDa considered as molecular switches, which cycle between two conformational states, an active state bound to GTP and an inactive state bound to GDP, where RhoA is the most studied member of this family [21]. ROCK, a downstream effector of RhoA, is a serine-threonine kinase of around 160 kDa, which in mammals is present in two isoforms, ROCK1 and ROCK2 [22,23]. The RhoA/ROCK pathway plays an important role in renal pathophysiology, where RhoA/ROCK participates in the regulation of pro-inflammatory cytokines such as TNF-α and IL-1β [19,24], increases the amount of transforming growth factor beta (TGF-β), and nuclear factor-κB (NFκB) [22]. On the other hand, the use of Fasudil, a selective ROCK inhibitor, has attracted great interest, as a potential tool in the prevention of kidney damage, in a wide variety of animal models including acute renal failure induced by ischemia-reperfusion, unilateral ureteral obstruction, hypertensive glomerulosclerosis, and renal failure induced by AngII [22]. Under experimental conditions, Fasudil has been shown to inhibit the cascade of events that leads to the increase in expression of pro-inflammatory cytokines, extracellular matrix genes, OS, and macrophage infiltration. Thus, both RhoA and ROCK have been considered as therapeutic targets to prevent kidney damage and hypertension [19,22,24,25]. However, it remains unknown how to predict when kidney damage can be prevented along the development of a chronic condition, which would not only fulfill our intellectual interest, but also could serve as a relevant tool to improve the prediction of Fasudil treatment effectiveness in clinic. 

The kidney regulates blood pressure mainly through excretion of Na^+^ and water, depending on the hormonal action of the RAS and other hormones with renal action [26]. However, in order to fulfill this function, the kidney requires the coordinated action of different cell types, including vascular and tubular cells [26]. The molecular mechanism of this intrarenal coordination has not yet been well established. A possible mechanism could involve cell-cell communication via gap junctions (GJs). In fact, connexins (Cxs), protein subunits of gap junctions (GJs), are present in the kidney, and direct communication between the different cell types of the nephron has been proposed to occur through GJs and/or hemichannels (HCs) [26]. In this sense, 9 Cx subtypes have been detected in the kidney (Cx26, Cx30.3, Cx31, Cx32, Cx37, Cx40, Cx43, Cx45 and Cx46), which are located in the vasculature or in different segments of the renal tubule, where most likely fulfill different physiological functions [26]. 

In cortical astrocytes, it has been demonstrated that two pro-inflammatory cytokines, TNF-α and IL-1β, reduce intercellular communication mediated by GJs and increase the permeability of the membrane through the HCs formed by Cx43 (Cx43 HCs) [27]. A similar change occurs in GJs and Cx43 HCs in astrocytes subjected to hypoxia-reoxygenation and in the presence of a high concentration of extracellular glucose [28] and in mesangial cells exposed to high glucose and pro-inflammatory cytokines [29]. This opposed regulation of GJs and Cx43 HCs by pro-inflammatory factors also occur in cultures of proximal tubule and mesangial cells treated with metabolic inhibitors or pro-inflammatory cytokines, where an increase in the activity of Cx HCs has been demonstrated [30,31]. In pathological conditions such as hypertension, the amount of renal Cxs is altered. For example, in the two kidneys-one clip model, an increase in the amount of Cx43 mRNA and protein was observed [32]. In addition, the amount of Cx43 is increased in inflammatory processes, in damaged renal tubules and in interstitial cells in human kidneys [33]. In three different models of CKD (i.e., the transgenic renin [Ren^+/+^] model, the administration of antibody against the glomerular basement membrane [α-GMB] and the unilateral obstructive uropathy), Toubas et al. observed an increase in the renal amounts of Cx43, so they postulated that the change in this Cx was caused by the development of inflammation in the damaged kidney [34]. Although renal tissue expresses several Cxs, only a few studies have described the involvement of GJs and HCs in kidney damage or in renal cell lines exposed to pathological condition and no signaling pathway has been clearly associated with these changes [30,31,34,35]. 

Therefore, their role in normal renal tissue or in the development and progression of kidney damage is largely unknown. Based on the evidence described above, in this study we evaluated whether the RhoA/ROCK pathway is activated and regulates the abundance of Cx43, which might contribute to renal tissue damage in the model of renal damage induced by infusion of AngII. 

## 2. Results

### 2.1. Four Weeks of Treatment with AngII Causes an Increase in Blood Pressure and Decrease in Renal Function

We first studied the time course of renal damage induced by AngII, as well as the spontaneous recovery of the kidney from the damage induced by AngII. For this purpose, we evaluated the damage in kidneys 2 weeks after removing the infusion pump from animals treated for 2, 3 or 4 weeks with AngII, (Figure 1A). Systolic blood pressure (SBP) was elevated in rats treated with AngII for 2, 3, 4, 5 and 6 weeks (in mmHg; AngII 2 = 170.0 ± 13.4, AngII 3 = 159.0 ± 18.1, AngII 4 = 192.0 ± 18.0, AngII 5 = 217.0 ± 3.0 and AngII 6 = 191.0 ± 12.0, respectively) used as positive control groups. Also, in rats treated with AngII for 4 weeks, and measured 2 weeks after removal of the pump, SBP remained high (AngII 4 + 2 = 160.0 ± 16.0 mmHg). In contrast, SBP returned to normal values (in mmHg; Ctrl 4 = 103.0 ± 6.0, Ctrl 5 = 95.0 ± 4.0 and Ctrl 6 = 103.0 ± 8.0) 2 weeks after withdrawing the stimulus in rats treated for 2 and 3 weeks with AngII (in mmHg; AngII 2 + 2 = 100.0 ± 6.0 and AngII 3 + 2 = 99.0 ± 4.0) (Figure 2A). 

To determine the degree of renal damage caused by the AngII treatment described above, the ratio urine protein/urine creatinine (UProt/UCrea) was measured. In rats treated with AngII for 2, 3, 4, 5 and 6 weeks this ratio was high (in arbitrary units, AU: AngII 2 = 20.6 ± 3.7, AngII 3 = 22.0 ± 7.9, AngII 4 = 50.0 ± 18.2, AngII 5 = 31.7 ± 10.2 and AngII 6 = 47.0 ± 5.4). However, in rats treated for 2 or 3 weeks and measured 2 weeks after stopping treatment with AngII, the ratio decreased (in AU: AngII 2 + 2 = 0.2 ± 0.1 and AngII 3 + 2 = 1.1 ± 0.1) to values similar to those of control rats (in AU: Ctrl 4 = 0.3 ± 0.2, Ctrl 5 = 0.3 ± 0.1 and Ctrl 6 = 0.4 ± 0.0). In contrast, in the group of rats treated with AngII for 4 weeks followed by 2 weeks without treatment, the ratio remained high (AngII 4 + 2 = 22.3 ± 9.5 AU) (Figure 2B).

### 2.2. The Suspension of AngII Infusion does not Reduce OS, Inflammation or Renal Tissue Damage in Rats Infused with AngII for 4 Weeks

The basic pathophysiological mechanisms of renal disorders are associated with redox imbalance and inflammatory response [11,36]. Ischemic or toxic phenomena that can damage the tubules, as well as the glomeruli, can be accompanied by excessive generation of ROS, and pro-inflammatory cytokines such as IL-1β and TNF-α [10,11,36]. In addition, in a wide range of renal diseases, macrophage infiltration (ED-1 positive cells) is closely related to the upregulation of tubular expression of osteopontin (OPN). OPN is a potent chemoattractant expressed by damaged kidneys and acts as an adhesion molecule for monocytes and macrophages [12,37]. Also, the development of interstitial fibrosis is thought to be the cause of the irreversibility of renal dysfunction, since myofibroblasts (Alpha-smooth muscle actin, [α-SMA] and collagen type III [Col III] positive cells) in the damaged renal tissue are the main cell effectors of renal fibrosis [38].

OS estimated through the concentration of TBARS in the supernatant of renal homogenates samples from rats treated with AngII during 2, 3, 4 and 6 weeks (in μmol/g; AngII 2 = 2.9 ± 0.2, AngII 3 = 2.9 ± 0.4, AngII 4 = 2.3 ± 0.3, and AngII 6 = 3.8 ± 0.3) was significantly higher than in samples of control kidneys (in μmol/g; Ctrl 4 = 1.3 ± 0.2, Ctrl 5 = 2.0 ± 0.1 and Ctrl 6 = 2.4 ± 0.2). This increase was not observed in samples from rats treated with AngII for 5 weeks (2.3 ± 0.2 μmol/g). The amounts of TBARS were similar to those found in kidneys obtained 2 weeks after stopping the AngII treatment in rats treated for 2, 3 and 4 weeks with AngII (in μmol/g; AngII 2 + 2 = 1.3 ± 0.2; AngII 3 + 2 = 2.0 ± 0.3 and AngII 4 + 2 = 2.4 ± 0.2) (Figure 3A). In addition, when this same parameter was measured in plasma samples from rats treated with AngII during 2, 3, 4, 5 and 6 weeks the amount of TBARS was higher (in μmol/L; AngII 2 = 3.5 ± 0.3, AngII 3 = 6.1 ± 0.6, AngII 4 = 9.9 ± 1.4, AngII 5 = 19.0 ± 1.9 and AngII 6 = 20.0 ± 4.1) compared to the value detected in the plasma of the respective control groups (in μmol/L; Ctrl 4 = 3.4 ± 0.4, Ctrl 5 = 10.0 ± 0.1 and Ctrl 6 = 4.7 ± 0.9). Two weeks after suspending the stimulus, the group of rats treated with AngII for 4 weeks still had high plasma TBARS concentration (AII 4 + 2 = 24.2 ± 3.8 μmol/L). However, 2 weeks after stopping the administration of AngII in rats treated for 2 and 3 weeks with AngII, the amount of TBARS in plasma decreased to values similar to those observed in control animals (in mmol/L; AngII 2 + 2 = 5.0 ± 1.5 and AII 3 + 2 = 6.6 ± 1.1) (Figure 3B).

The immunostaining for ED-1 showed a higher intensity in the group of animals treated for 4 weeks with AngII and was even greater in the groups treated for 5 and 6 weeks, compared with the control rats. After 2 weeks of AngII withdrawal, the reactivity for ED-1 decreased in the rats treated for 2 and 3 weeks, reaching an intensity similar to that of kidneys from control rats. In contrast, in kidneys from rats treated with AngII for 4 weeks, the reactivity for ED-1 was still high 2 weeks after AngII was suspended (Figure 4A). 

The response of IL-1β to the treatment was slightly different. After 2 weeks of stopping the AngII treatment, the amounts of IL-1β in the homogenate decreased (AngII 2 + 2 = 206.0 ± 3.0 pg/g protein) to values similar to those found in the kidney of control rats (in pg/g protein; Ctrl 4 = 179.0 ± 20.0, Ctrl 5 = 185.0 ± 20.0 and Ctrl 6 = 203.0 ± 38.0) only in rats treated for 2 weeks with AngII. Interestingly, in kidneys from rats that were treated for 3 and 4 weeks with AngII, the amounts of IL-1β remained significantly elevated (in pg/g protein; AngII 3 + 2 = 415.0 ± 26.0 and AngII 4 + 2 = 350.0 ± 12.0) (Figure 4B).

The degree of fibrosis, studied by immune detection of Col III, revealed more reactivity in kidney sections from rats treated with AngII during 2, 3, 4, 5 and 6 weeks, compared with that found in kidneys of control rats. However, 2 weeks after AngII withdrawal, the immunostaining for Col III decreased in kidneys of rats that had received 2 or 3 weeks of the AngII treatment to values similar to those found in kidneys of control rats, but the group treated for 4 weeks with AngII maintained high renal immunoreactivity for Col III, 2 weeks after suspending the AngII administration (Figure 5). 

### 2.3. The Increase in Cx43 and Activation of RhoA/ROCK Become Independent of the Stimulus after 4 Weeks of Treatment with AngII

The phosphorylation state of renal myosin phosphatase target subunit 1 (MYPT-1), a downstream target of ROCK, in the group of rats treated with AngII during 4, 5 and 6 weeks was significantly higher (in AU: AngII 4 = 1.0 ± 0.1, AngII 5 = 0.9 ± 0.2 and AngII 6 = 1.1 ± 0.3) than in control rats. Two weeks after stopping treatment with AngII, the state of phosphorylation of MYPT-1 in kidneys from rats treated for 2 and 3 weeks with AngII (in AU: AngII 2 + 2 = 0.2 ± 0.0 and AngII 3 + 2 = 0.2 ± 0.0) was similar to control values (in AU: Ctrl 4 = 0.3 ± 0.1, Ctrl 5 = 0.2 ± 0.1 and Ctrl 6 = 0.2 ± 0.1). In contrast, in the kidney samples from the group of rats treated for 4 weeks with AngII, the phosphorylation status of MYPT-1 remained high (1.0 ± 0.1 AU) (Figure 6A). 

Similarly, the amount of Cx43 was higher in the kidneys from rats treated with AngII during 4, 5 and 6 weeks (in AU: AngII 4 = 1.7 ± 0.1, AngII 5 = 1.6 ± 0.2 and AngII 6 = 1.3 ± 0.1) compared to the kidneys from control rats. Two weeks after stopping treatment with AngII, the amount of Cx43 in the kidneys of animals treated for 2 or 3 weeks with AngII decreased (in UA: AII 2 + 2 = 0.3 ± 0.1 and AngII 3 + 2 = 0.4 ± 0.1) to values similar to those detected in kidneys from control rats (in AU: Ctrl 4 = 0.5 ± 0.1, Ctrl 5 = 0.3 ± 0.0 and Ctrl 6 = 0.5 ± 0.1). However, 2 weeks after stopping the treatment in rats treated with AngII for 4 weeks, the amount of Cx43 remained high (AngII 4 + 2 = 1.2 ± 0.1 AU) (Figure 6B).

### 2.4. Fasudil Prevents Kidney Damage, OS, Inflammation, Fibrosis, and the Increase in Protein Amount of Cx43, but does not Decrease SBP in Rats Treated with AngII

After having identified 4 weeks of treatment with AngII as a point at which renal damage is not spontaneously reversible, we evaluated the role of Rho/RACK on organism and organ parameters. For this purpose, we used four groups of animals: two control groups (Ctrl and Ctrl + Fasudil), and two experimental groups (AngII administered for 6 weeks and AngII + Fasudil, administered for the last 4 weeks of a 6 weeks treatment with AngII) were studied. Fasudil (100 mg/kg/day) was given in the drinking water (Figure 1B). The SBP in rats treated only with AngII (192.0 ± 11.0 mmHg) and in rats treated with AngII plus Fasudil (AngII + Fasudil, 202.0 ± 10.0 mmHg) were significantly higher than those values recorded in their respective controls (Ctrl and Ctrl + Fasudil, in mmHg: 103.0 ± 8.0 and 97.0 ± 3.0, respectively). Therefore, Fasudil did not reverse the increase in SBP induced by AngII (Figure 7A). 

The ratio U Prot/UCrea increased significantly in rats treated with AngII (73.0 ± 14.2 AU) compared with those of the control group (in AU: Ctrl = 0.4 ± 0.2, Ctrl + Fasudil = 0.4 ± 0.2). However, in rats treated with AngII + Fasudil, these values were similar to those of the control rats (1.9 ± 0.6 AU) (Figure 7B). In Table 1, the individual values for each parameter measured to assess the renal function of these rats are shown. A similar pattern was observed in proteinuria, fractional excretion of Na^+^ and K^+^ and creatinine clearance. 

Consistent with previous results, the amount of TBARS (Figure 7C and D), the distribution of ED-1 (Figure 7E) and the amount of IL-1β (Figure 7F), Col III immunoreactivity (Fig. 7G), amounts of p-MYPT-1 (Figure 7H) and Cx43 (Figure 7I) in the group treated with AngII for 6 weeks were high, but in kidneys from rats treated with AngII + Fasudil, the values for these parameters were similar to that detected in the kidneys of control rats.

## 3. Discussion

The incidence of CKD is increasing worldwide, and there is no efficient therapy to reduce this phenomenon. The main therapies currently available focus on the control of blood pressure and the optimization of the blockade of the RAS [39]. In the present work, we addressed two questions. First, considering that AngII produces damage in the kidney, we identified the time point when renal damage turns irreversible, which resulted in be after 4 weeks of treatment with AngII since SBP, inflammation, OS, fibrosis, the amount of Cx43 and p-MYPT-1 remained high even after 2 weeks of AngII withdrawal. This could be explained by renal cell dysfunction that develops by the initial action of AngII and establishes interconnections between OS, inflammation, and fibrosis [40]. Although The half-life of bioactive peptides such as AngII is likely to be about a few hours [41], these interconnections coexist and communicate with each other independent of AngII, thereby exacerbating the processes underpinning these different entities with the end result of high morbidity and mortality in this model of kidney damage [40]. Also, we found an increase in kidney macrophage infiltration (ED-1) and renal amounts of IL-1β in response to AngII, indicating the presence of an inflammatory response mediated by the innate immune system. Interestingly, the amounts of IL-1β also remained elevated even when the stimulus was withdrawn. These findings suggest that kidney cells remain inflamed independent of the absence of cells of the innate immune system. This possibility is supported by the known activation of kidney inflammasome upon different injuries [42].

α-SMA and Col III also showed a similar response pattern. Accordingly, Ozawa et al. described that rats stimulated with AngII for 6 days and then 6 days without the stimulus, presented interstitial fibrosis, glomerular hypertrophy, macrophage infiltration, increase in MCP-1 and also in mRNA quantity of transforming growth factor-β (TGF-β) [8], showing a persistence of the effect even in the absence of the original stimulus. It is also known that TGF-β stimulates the synthesis of extracellular matrix and inhibits the action of proteases that degrade the matrix [9,43], favoring the development of fibrosis and inflammation in kidneys, finally reducing function and increasing kidney damage [9,12,44]. 

On the contrary, these parameters were spontaneously reversed in animals infused with AngII for 3 or less weeks, which indicates that AngII can generate alterations that can be compensated by kidney tissue that was not affected by AngII and/or can recover thanks to the small regeneration capacity of kidney tissue [45]. Spontaneous recovery has also been found by Frenay et al., who showed that newly formed renal lymph vessels can undergo spontaneous regression after cessation of AngII [45], suggesting that lymph vessel formation is a key event in renal interstitial fibrosis [46]. In the kidney, lymph angiogenesis occurs after proteinuria is established and prior to collagen deposition, fibrosis and macrophage infiltration, whereas angiotensin converting enzyme (ACE) inhibition significantly prevents this new lymph vessel formation [45,47]. Papakrivopoulou et al. by analyzing mouse models of irreversible and reversible glomerular injury, described that Van Gogh-like 2 (Vangl2), a protein that regulates planar cell polarity (PCP) is implicated in acquired kidney disease by modulating tissue remodeling that includes cytoskeletal rearrangements affecting cell morphology or changes in inflammation [48]. Indeed, downregulation of Vangl2 increased the activity of matrix metalloproteinase-9 (MMP9). MMP9 has previously been shown to be produced by podocytes, and it is altered in several glomerular diseases, supporting the conclusion that Vangl2 modulates glomerular injury by stopping MMP9 production [48]. Therefore, it is possible that TGF-β, lymph angiogenesis and/or Vangl2 could be involved in the development of renal fibrosis and inflammation, and thus it could be a key factor in the irreversibility of kidney damage; consequently, additional studies are required to address and demonstrate the involvement of these factors in the reversibility/irreversibility of renal damage, independent of the initial stimulus. 

We were also interested in predicting the effectiveness of Fasudil to reduce renal damage. For this reason, we directed our investigation to the RhoA/ROCK pathway. In this sense, the renal afferent arterioles are primarily responsible for regulating preglomerular resistance, renal blood flow, and GFR. Elevated renal vascular resistance and preglomerular reactivity are observed in AngII-induced hypertension [49]. Although many systemic, neural, paracrine, and autoregulatory mechanisms contribute to afferent arteriolar dynamics, in AngII-dependent hypertension a direct effect was observed between the RhoA/ROCK pathway and the endogenous production of AngII [49]. We observed that, although treatment with Fasudil did not reduce SBP, the establishment of irreversible renal damage was prevented, reducing inflammation, OS, fibrosis, and also kept the amount of Cx43 and p-MYPT-1 at normal levels. Interestingly, the activity of the RhoA/ROCK pathway has been widely investigated in the pathogenesis of hypertension. Under this condition, this pathway fulfills one of the important roles in regulating smooth muscle contraction, could lead to an increase in peripheral vascular resistance, which are characteristics observed in several hypertension models [50]. The protective effect of Fasudil in vivo is partly explained by its pleiotropic action in different systems; therefore, considering that ROCK inhibitors were developed as anti-hypertensive drugs, it is striking that in the present work Fasudil did not affect SBP, but did reduce the progression of kidney damage. Against the effect of this drug on blood pressure, several studies have established that Fasudil is renoprotective without affecting blood pressure, establishing a controversy regarding the use of Fasudil and its anti-hypertensive action [51,52,53,54]. Because ROCK is involved in various cellular functions, the renoprotective effect of Fasudil appears to be mediated by multiple mechanisms, such as the inhibition of extracellular matrix genes, the reduction of macrophage/monocyte infiltration and the reduction of NADPH oxidase subunits gene expression, all of which are not directly related to blood pressure [55]. Although it has been demonstrated that Fasudil has several beneficial effects, there are many limitations in clinical use, including short-course treatment, low oral bioavailability, cell toxicity and blood pressure fluctuations [56]. In view of this, it is possible that tissues different from kidney take the lead in the hypertensive response. Similar interpretation might explain the persistent high TBARS concentration in plasma and evident recovery in rat kidney function subjected to the protocol AngII 4 + 2 (Figure 3B). Thus, the circulating TBARS in these rats might be generated in other tissues affected by the high SBP resistant to Fasudil. However, further studies remain to be done to elucidate why Fasudil does not prevent the increase in SBP, even when it prevents kidney damage.

Other authors have shown that in afferent arteriolar cells from rats treated with AngII, the activation of NF-κB is mediated by the RhoA/ROCK pathway and the ROCK/NF-κB axis contributes to the upregulation of angiotensinogen, leading to an increase in the amount of intrarenal AngII [49]. In addition, in fibroblasts, a direct relationship has been demonstrated between the activation of the RhoA/ROCK pathway and the increase in the amount of Cx43. In these cells, the expansion mechanisms in response to stretching, involves the release of ATP to the extracellular medium through the RhoA/ROCK pathway and the activation of Cx HCs. In addition, the treatment with Y-27632 (another ROCK inhibitor) or with blockers of Cx HCs, such as octanol or carbenoxolone, inhibited the increase of ATP in the extracellular medium and the growth of fibroblasts [57]. Xie et al. explored the mechanism for the reduction in Cx43 levels induced by RhoA/ROCK signaling in high glucose-treated glomerular mesangial cells (GMCs) [58]. Their results indicated that activated RhoA/ROCK signaling induced Cx43 degradation in GMCs cultured in high glucose depending on F-actin regulation that promoted the association between ZO-1 and Cx43, which possibly triggered Cx43 endocytosis [58]. However, it is important to also consider that in a previous work, we demonstrated that in the glomerular mesangial cell line (MES-13), AngII promotes a feedforward mechanism in which three non-selective channels (Cx HCs, pannexin1 channels and purinergic receptor P2 × _7_ receptors) maintain or even amplify inflammatory and oxidative responses, causing damage to kidney cells [31]. In this work, AngII-induced alterations in cell membrane permeability could lead to activation of several metabolic pathways including the RhoA/ROCK signaling pathway by the progressive increase in the amount of p-MYPT in MES-13 cells, which promote OS and generation of pro-inflammatory cytokines [31]. Interestingly, we found similar changes in RhoA/ROCK activity and also found that ROCK inhibitors, prevented increases in the amount of Cx43 induced by AngII, indicating that activation of a RhoA/ROCK-dependent pathway and Cx43 HCs are regulated by the same transduction mechanism and intracellular signaling pathway activated by AngII. According to the results obtained in this work, we postulate that there is a direct correlation between the activity of a RhoA/ROCK-dependent pathway, Cx43 and renal damage in this model of hypertensive nephropathy. 

In conclusion, the rat renal damage, inflammation, OS, changes in amount of Cx43 and activity of the RhoA/ROCK pathway are spontaneously reversible if the increase in AngII lasts 3 weeks or less. These alterations become irreversible and irreparable if AngII administration is maintained for 4 or more weeks. Fasudil, an inhibitor of the RhoA/ROCK pathway, prevents the irreversibility of the evaluated kidney alterations when is administered 2 weeks before the point of irreversibility revealing a close relationship between activation of a RhoA/ROCK-dependent pathway and increase in the amount of Cx43 in CKD (Figure 8). This would not only fulfill our intellectual interest but also could serve as a relevant tool to improve the prediction of Fasudil treatment effectiveness in CKD in a clinical context. The present results obtained with the least invasive measurements suggest that elevated SBP and UProt/UCrea with normal plasma TBARS would predict that Fasudil treatment might prevent the establishment of CKD induced by high AngII concentration.

## 4. Materials and Methods

### 4.1. Reagents and Antibodies

Triton X-100, Tris-HCl, hydrogen peroxide, malondialdehyde tetrabutylammonium salt (MDA), λ-carrageenan, Fasudil, phenylmethanesulfonyl fluoride (PMSF), 3.3′-diaminobenzidine, phosphate salts, other chemicals and the antibodies anti-α-tubulin and anti-collagen type III (Col III) were purchased from Sigma-Aldrich (St. Louis, MO, USA). Monoclonal antibody against macrophages (clone ED-1) was obtained from AbD Serotec (Kidlington, UK). The monoclonal anti-MYPT1 antibody was obtained from BD Transduction Laboratories (San José, CA, USA) and the polyclonal anti-phosphorilated-MYPT1 (Thr696) antibody was obtained from Merck Millipore (Darmstadt, Germany). the monoclonal anti-unphosphorylated Cx43 antibody was obtained from Invitrogen (Carlsbad, CA, USA). Secondary antibodies and PAP complexes were obtained from ICN Pharmaceuticals-Cappel (Costa Mesa, CA, USA) or from Santa Cruz Biotechnology Inc. (Santa Cruz, CA, USA).

### 4.2. Animals

Male Sprague–Dawley 1 month old rats (100–120 g) were kept in a 12 h light/12 h dark cycle, with water and food (5% fiber, 20.5% protein, 4% fat; Champion, Santiago, Chile) ad libitum at the university animal care facilities. Animals were placed in metabolic cages for 16 h to collect urine in a container built into the cage. Urine was measured and aliquoted After urine collection, animals were anesthetized with ketamine/xylazine (10:1 mg/kg of body weight, ip). The abdominal aorta was used to collect blood samples. Blood was then centrifuged, and plasma was frozen for further analysis. Kidneys were processed for Western blotting and immunohistochemistry. Animals were sacrificed by exsanguination under anesthesia. All procedures were in accordance with institutional and international standards for the humane care and use of laboratory animals (Animal Welfare Assurance Publication A5427-01, Office for Protection from Research Risks, Division of Animal Welfare, NIH (National Institutes of Health), Bethesda, MD, USA), as described previously [59]. The Bioethical and Biosafety Committee of the Faculty of Biological Sciences at the Pontificia Universidad Católica de Chile (CBB-064/2014; 13 June 2014) approved the described experimental procedures. 

### 4.3. AngII Administration and Experimental Procedure

We used the previously described hypertensive model induced by the prolonged administration of AngII [60]. Briefly, animals were anesthetized with a dose of ketamine/xylazine (25:2.5 mg/kg, respectively). A small (1 cm) cut was made in the mid scapular region and an osmotic pump was implanted (Alzet, Cupertino, CA, USA), and subsequently the wound was sutured. Pumps were loaded with AngII dissolved in saline with a release rate of 160 ng/min for 6 weeks [60,61]. Pumps were kept in the animals for the duration of the experimental treatment. The Sham rats (controls) underwent the same surgical procedure, but without implanting the osmotic pump. After their recovery, the rats were maintained with water and food ad libitum. To check that the implanted rats did indeed develop high blood pressure, systolic blood pressure (SBP) was measured weekly. In the first set of experiments, we evaluated the spontaneous reversibility of the renal damage with AngII. For this purpose, rats were treated for 2, 3 or 4 weeks with AngII, and then the pump was removed, and the animals were evaluated for 2 additional weeks. As positive controls for these groups, rats kept with AngII for the whole period (2, 3, 4, 5 or 6 weeks) were used. As a negative control, we used rats that were never exposed to AngII (Figure 1A). In the second set of experiments, 2 control groups (Ctrl and Ctrl + Fasudil), and 2 experimental groups (AngII administered for 6 weeks and AngII + Fasudil administered for the last 4 weeks) were studied. Fasudil (100 mg/kg/day) was given in the drinking water (Figure 1B).

### 4.4. Blood Pressure Measurements

SBP was determined once a week, in the morning, in conscious pre-warmed restrained rats by non-invasive plethysmography (NIBP machine, IITC Inc., Woodland Hills, CA, USA) by the tail-cuff method. At least four determinations were made in every session and the mean of the four measurements was taken as the SBP value.

### 4.5. Renal Function Measurements

Plasma and urinary creatinine levels were measured by the Jaffé alkaline picrate assay (VALTEK Diagnostica, Santiago, Chile). Urinary protein concentration was determined by Bradford’s method (Bio-Rad protein assay, Kidlington, UK) [62]. Creatinine clearance was calculated according to the standard formula C = (U × Ṽ/P), where C is the creatinine clearance, U is the creatinine urinary concentration, Ṽ is the urine flow rate per minute, and P is the creatinine plasmatic concentration [59]. In the samples of urine and plasma, the amounts of Na^+^ and K^+^ were measured using the electrolyte meter 9180 Electrolyte Analyzer (Roche). To determine the fractional excretion of Na^+^ and K^+^, the following equation was applied:$$\mathrm{FE}\;{\mathrm{Na}}^{+}\;\mathrm{or}\;K^{+}=\frac{\left(\mathrm{PC}*\left[{\mathrm{Na}}^{+}\right]U\;\mathrm{or}\;\left[K^{+}\right]U\right)}{\left(\mathrm{CU}*\left[{\mathrm{Na}}^{+}\right]P\;\mathrm{or}\;\left[K^{+}\right]P\right)}*100$$where, [Na^+^]U or [K^+^]U = Concentration of Na^+^ or K^+^ in urine; and [Na^+^]P or [K^+^]P = Concentration of Na^+^ or K^+^ in plasma. PC = Plasma creatinine and CU = Creatinine in urine.

To determine the ratio U prot/U Crea, the value for urinary protein was divided by the value for urinary creatinine.

### 4.6. Thiobarbituric Acid Reactive Substances (TBARS) Measurement

A modified version of the method published by Ramanathan and collaborators [63] was used to measure TBARS. Briefly, supernatant of renal homogenates or plasma were mixed with thiobarbituric acid (0.8% TBA w/v), SDS (8% w/v) and acetic acid (20% v/v) and heated for 60 min at 90 °C. Afterwards, precipitated material was removed by centrifugation, and absorbance was measured at 532 nm. MDA was used as standard for the calibration curve to estimate TBARS.

### 4.7. Histological Damage Assessment

A semi-quantitative morphometric analysis was used to determine tissue damage. Breafly, a score on a scale from negative to positive three (− to +++), was defined according to the immunoreactivity intensity observed in the kidney tissue sample for each of the antibodies used (OPN, ED-1, α-SMA, Col III), from ([−] = 0–10%, [+] = 10–40%, [++] = 40–70% y [+++] = 70–100% of the area observed) [64,65,66,67]. To assess the degree of fibrosis, staining of collagen fibrils by Sirius red F3BA was carried out as previously described [68].

### 4.8. Tissue Processing and Immunohistochemical Analysis

Slices of 5 mmincluding cortex, medulla, and papilla, were made from kidneys and were fixed by immersion in Bouin’s solution for 24 h at room temperature, followed by dehydration, and embedding in Paraplast (Monoject Scientific, St. Louis, MO, USA). Sections of 5-mm thickness were made with a rotatory microtome, mounted on glass slides, and stored until immunostaining. An indirect immunoperoxidase technique was used to perform the mmunolocalization studies [69]. To this purpose, sections were dewaxed, rehydrated, equilibrated in immune solution (IS) (in M: 0.11 Na_2_HPO_4_, 0.04 KH_2_PO_4_, 1 NaCl, 0.32 Tris-HCl, and 0.03 sodium azide) pH 7.6, and incubated with the primary antibody (1:100) overnight at room temperature. The next day, sections were washed, and incubated for 30 min with the corresponding secondary antibody (1:20) followed by the incubation with a peroxidase-anti-peroxidase (PAP) complex (1:150). Immunoreactivity was observed using a 0.1% (w/v) 3.3′-diaminobenzidine and 0.03% (v/v) hydrogen peroxide solution. The sections were, counterstained with hematoxylin, dehydrated, and coverslipped. A Nikon Eclipse E600 microscope and Nikon DXM1200 digital camera were used to acquire the images, as previously described [70]. The stained areas in each image were quantified with a computer-assisted image-analysis software (Simple PCI, Hamamatsu, Japan). An average of the total immunostained cells was expressed as the mean absolute values per square micron, as previously described [70].

### 4.9. Enzyme-linked Immunosorbent Assay (ELISA)

ELISA assays were performed to determine the amount of IL-1β in the kidney. Rats were anesthetized with ketamine/xylazine (25:2.5 mg/kg of body weight, i.p.). Afterwards the kidney was removed and homogenized with an Ultra-Turrax homogenizer in buffer (ratio: 0.1 g renal tissue: 1 ml lysis buffer) containing 100 mM Tris-HCl pH 7.4, 5 mM EDTA, 1% SDS, 1 μM PMSF and a protease inhibitor cocktail (Halt™ Protease Inhibitor Cocktail, Pierce, Rockford, IL, USA). Then, samples were centrifuged at 14,000× *g* for 10 min (Eppendorf 5415C, Eppendorf, Hamburg, Germany). Supernatants were collected and protein content assayed by Bicinchoninic Acid (BCA) Protein Assay. Cytokine levels were determined by sandwich ELISA, according to the manufacturer’s protocol (IL-1β EIA kit, Enzo Life Science, USA). For the assay, 100 µL of samples were added per well and incubated at 4 °C overnight. A calibration curve with recombinant cytokine was included. Samples were incubated with secondary antibody at room temperature for 2 h and the reaction was developed with avidin–HRP and substrate solution. Absorbance was measured at 450 nm with reference to 570 nm with the microplate reader Synergy HT (Biotek Instruments). The results were normalized by protein amount in pg/g protein.

### 4.10. Western Blot Assays

Proteins from renal tissue samples (50 μg of proteins) were resolved by electrophoresis in 10% SDS-polyacrylamide gel, using one lane for the pre-stained molecular weight markers. Proteins were transferred to a PVDF membrane (pore size: 0.45 μm), blocked at room temperature with Tris pH 7.4, 5% skim milk (w/v) and 1% BSA (w/v). Then, the PVDF membrane was incubated overnight at 4 °C with anti-Cx43 (1:1,000), anti-p-MYPT1 (1:500), or anti-MYPT1 (1:1,000) antibody, followed by incubation with secondary antibody conjugated to peroxidase (1:2,000) for 1 h at room temperature. Then, the PVDF membrane was stripped and reblotted with the anti-α-tubulin antibody (1:5,000) used as loading control, following the same procedure described above. After repeated rinses, immunoreactive proteins were detected with ECL reagents (Pierce Biotechnology, Rockford, IL, USA) according to the manufacturer’s instructions. The bands detected were digitized and subjected to densitometry analysis using the software Image J (Version 1.50i, NIH, Washington, DC, USA).

### 4.11. Statistical Analysis

Results were expressed as mean ± standard error (SE); *n* is the number of independent experiments. Figure legends included detailed statistical results. GraphPad Prism (version 7, GraphPad Software, La Jolla, CA) was used to perform statistical analyses. One or two-way analysis of variance (ANOVA) was performed for multiple comparisons When significance was obtained, a Tukey’s post-hoc test was performed. A probability of *p* < 0.05 was considered statistically significant. Results are expressed as the average of values from each independent experiment ± SE.

## Figures and Tables

**Figure 1 ijms-20-04408-f001:**
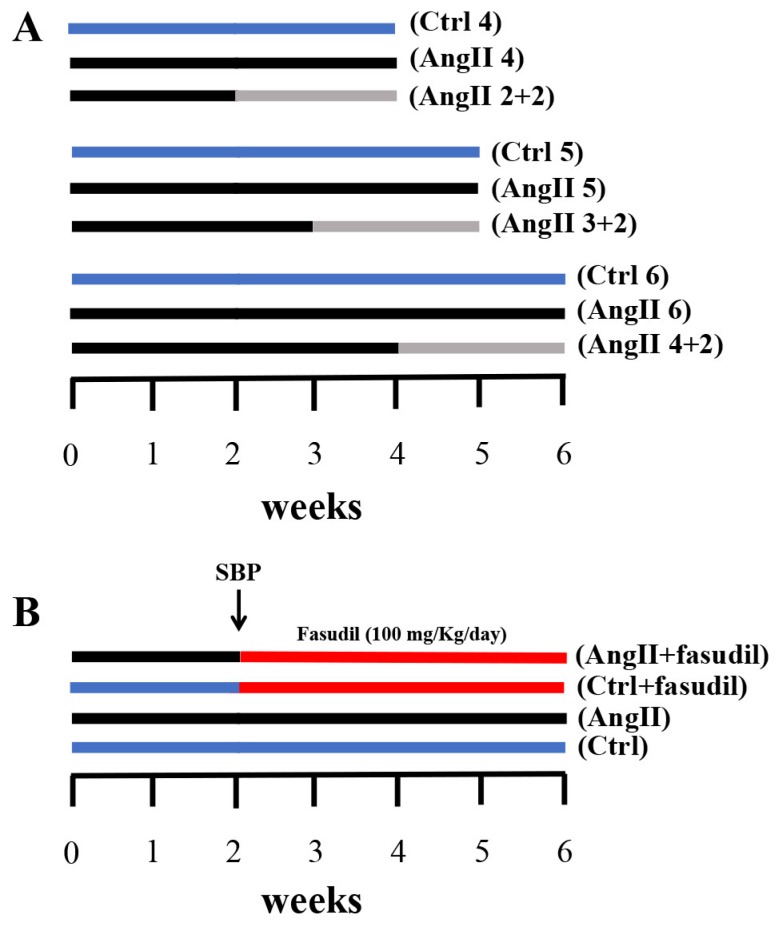
Outlines of experimental protocols developed in male Sprague Dawley rats. (**A**) We used three experimental groups that were evaluated at 4, 5 and 6 weeks, respectively. Each group was divided into three subgroups: one control Ctrl (blue line); another with AngII infusion (160 ng/min, black line) during 4, 5 and 6 weeks (AngII 4, AngII 5 and AngII 6); and a third one in which animals were treated with AngII for 2, 3 and 4 weeks and then the treatment was suspended for the next 2 weeks (AngII 2 + 2, AngII 3 + 2 and AngII 4 + 2, grey line). (**B**) Four experimental groups were used; these were divided into two control groups and two groups with AngII administration for 6 weeks. At 2 weeks after the initiation of the administration of AngII, systolic blood pressure (SBP) was measured and the groups were subdivided into rats treated with Fasudil (100 mg/kg/day) (AngII + Fasudil) (Ctrl + Fasudil) for 4 weeks (red line) and rats that continued to be treated with only AngII until completing 6 weeks (AngII) and their controls (Ctrl).

**Figure 2 ijms-20-04408-f002:**
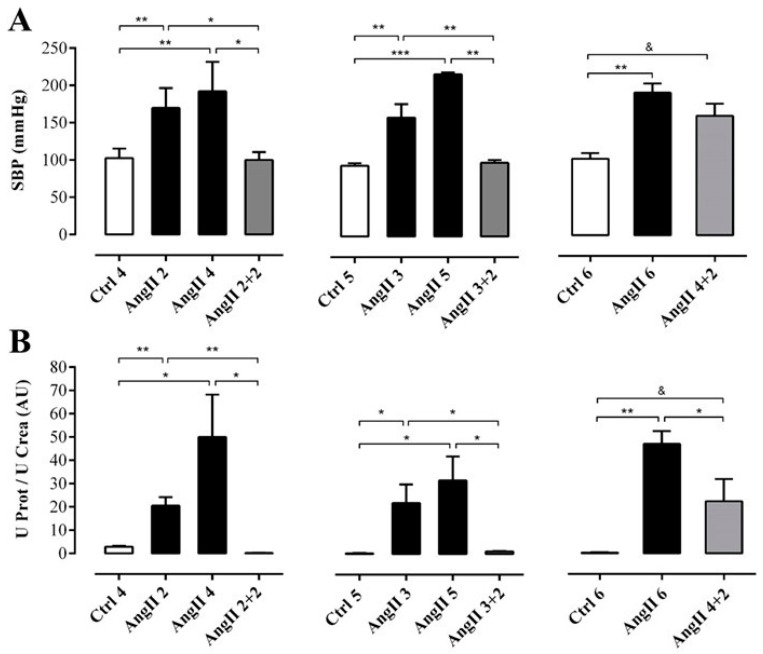
While the SBP remained elevated, the renal function had partly recovered in rats after 2 weeks of withdrawal from 4 weeks treatment with AngII. SBP (mmHg) was measured by plethysmography (**A**). Protein and creatinine were measured in urine samples to assess renal function from ratio UProt/UCrea (**B**). The bars represent the means ± SE of a *n* ≥ 4 rats per experimental group. The differences between the subgroups in each of the three groups were evaluated by an ANOVA followed by a Tuckey test. *** *p* < 0.001, ** *p* < 0.01 and * *p* < 0.05 vs. AngII groups; & *p* < 0.05 vs. AngII 4 + 2.

**Figure 3 ijms-20-04408-f003:**
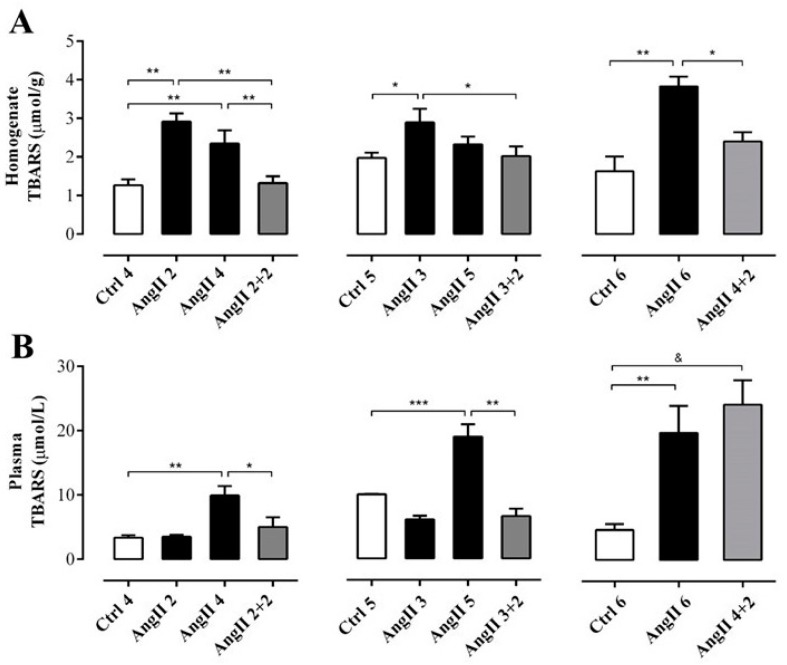
The amount of TBARS remains high in the plasma, but not in the kidney of rats treated with AngII for 4 weeks. Supernatant of kidney homogenates (**A**) and plasma (**B**) were used to measure TBARS as marker of oxidative state. The bars represent the means ± SE of a *n* ≥ 4 rats per experimental group. The differences between the subgroups of each of the three groups were evaluated by an ANOVA followed by a Tuckey test. *** *p* < 0.001, ** *p* < 0.01 and * *p* < 0.05 vs. AngII groups; & *p* < 0.05 vs. AngII 4 + 2.

**Figure 4 ijms-20-04408-f004:**
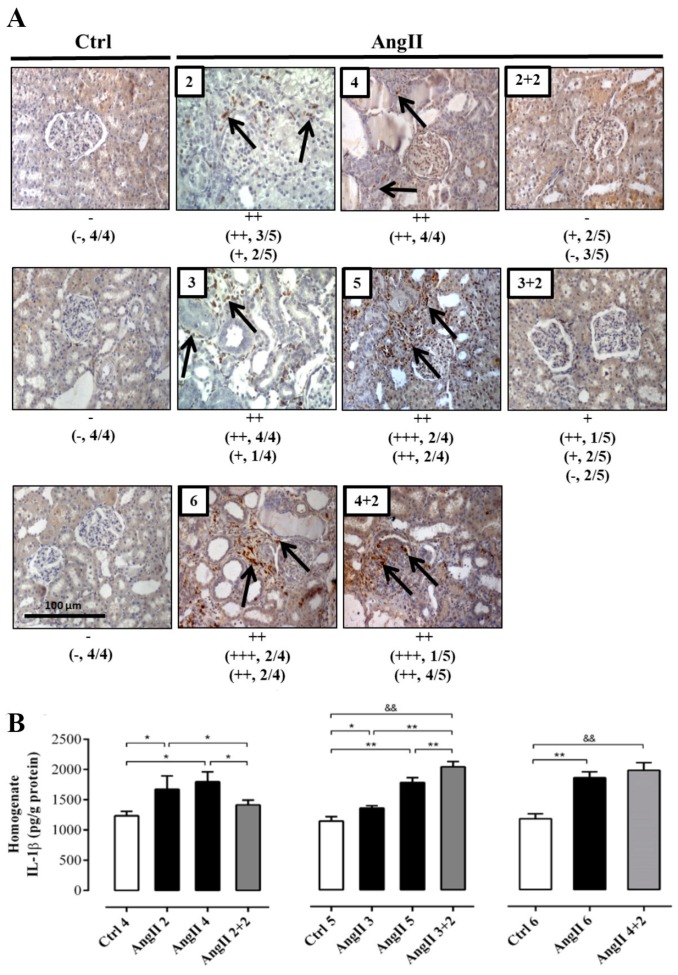
The amounts of ED-1 and IL-1β remain high in rats treated with AngII for 4 weeks. Immunostaining for ED-1 were performed in renal samples (**A**). Representative pictures of at least three different kidneys and four different fields are shown. Scale bar = 100 μm. Brown staining represents ED-1 positive immunoreactivity. The staining is indicated by black arrows. The semi-quantitative morphometric analysis for ED-1 was performed as explained in the Materials and Methods. Renal homogenates were used to measure IL-1β as marker of inflammation (**B**). The bars represent the means ± SE of a *n* ≥ 4 rats per experimental group. The differences between the subgroups of each of the three groups were evaluated by an ANOVA followed by a Tuckey test. ** *p* < 0.01 and * *p* < 0.05 vs. AngII groups; && *p* < 0.01 vs. AngII 3 + 2 and AngII 4 + 2.

**Figure 5 ijms-20-04408-f005:**
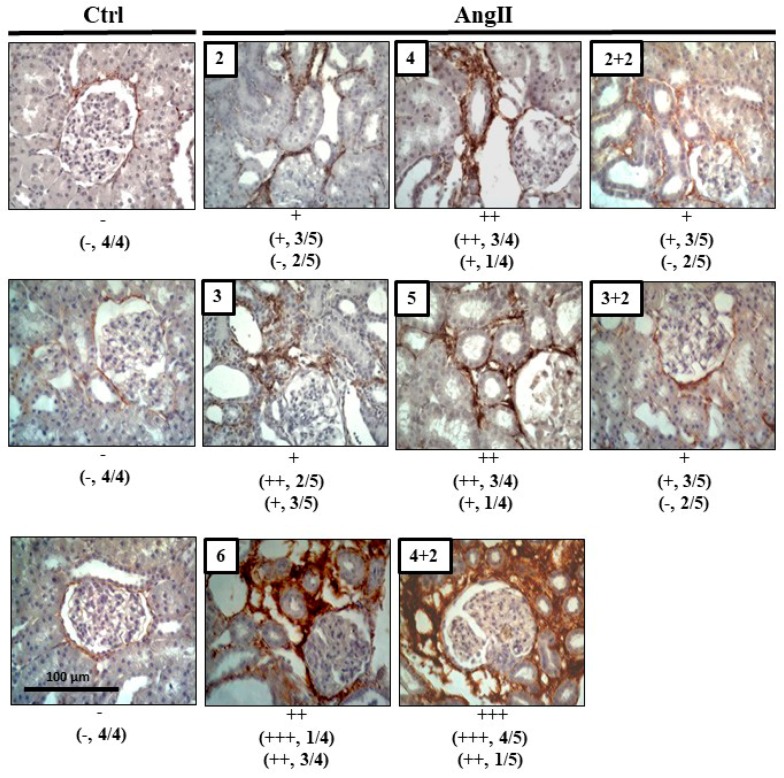
Col III remains high after 4 weeks of infusion with AngII. Immunostaining for Col III was performed in renal samples. Representative pictures of at least three different kidneys and four different fields are shown. Scale bar = 100 μm. Brown staining represents the marker for Col III. The semi-quantitative morphometric analysis for Col III was performed as explained in Materials and Methods.

**Figure 6 ijms-20-04408-f006:**
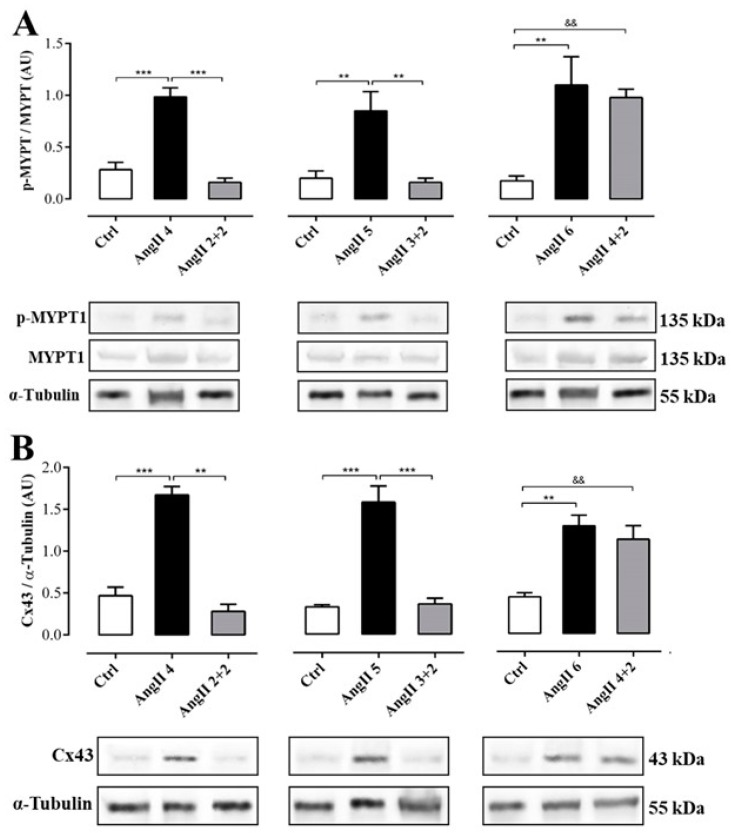
The amounts of phosphorylated MYPT and Cx43 was elevated in rats treated with AngII for 4 weeks. Graphs show MYPT-1 phosphorylation (**A**) and Cx43 relative amounts (**B**). Representative pictures of MYPT (phosphorylated p-MYPT and unphosphorylated MYPT), and Cx43 positive bands and α-tubulin as loading control () are shown under the graph. Bars represent the means ± SE of a *n* ≥ 4 animals in each experimental group. The differences between subgroups were evaluated by an ANOVA followed by a Tuckey test. *** *p* < 0.001, ** *p* < 0.01 vs. AngII groups; && *p* < 0.01 vs. AngII 4 + 2.

**Figure 7 ijms-20-04408-f007:**
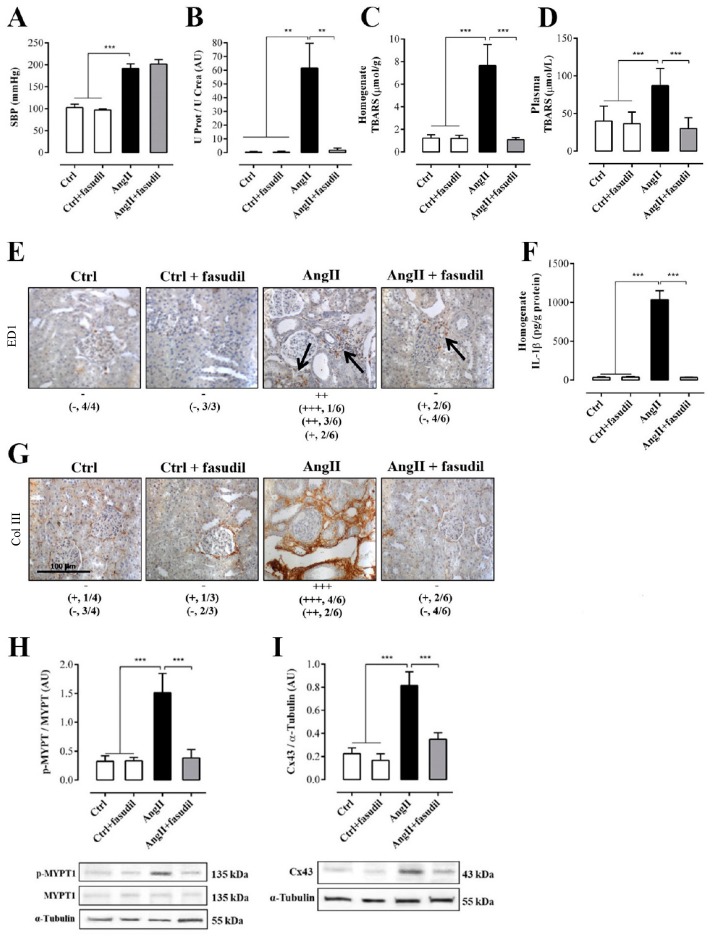
Fasudil prevents kidney damage, OS, inflammation, fibrosis, and the increase in protein amount of Cx43, but does not decrease SBP in rats treated with AngII (**A**). Protein and creatinine were measured in urine samples to assess renal function from ratio UProt/UCrea (**B**). Supernatants of kidney homogenates (**C**) and plasma (**D**) were used to measure TBARS as a marker for the oxidative state. Immunostaining of ED-1 was performed in renal samples (**E**), and kidney homogenates were used to measure levels of IL-1β as marker for inflammation (**F**). Immunostaining of Col III was performed in renal samples (**G**). Representative pictures of at least three different kidneys and four different fields are shown. Scale bar = 100 μm. Brown staining represents ED-1 and Col III positive immunoreactivity. The semi-quantitative morphometric analysis for ED-1 and Col III was performed as explained in Materials and Methods. Graphs show relative amount of phosphorylated MYPT-1 (p-MYPT) (**H**) and the relative amount of Cx43 (**I**). Under the graph representative pictures of p-MYPT, unphosphorylated MYPT and Cx43 positive bands and its loading control (α-tubulin) are shown. The bars represent the means ± SE of a *n* ≥ 4 rats per experimental group. The differences between the subgroups of each of the three groups were evaluated by an ANOVA followed by a Tukey test. *** *p* < 0.001, ** *p* < 0.01 and * *p* < 0.05 vs. AngII group.

**Figure 8 ijms-20-04408-f008:**
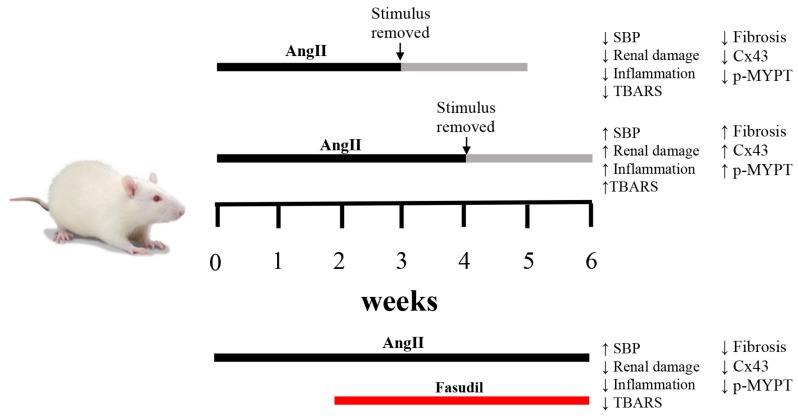
Schematic representation of temporal changes in response to AngII and Fasudil. When rats are treated with AngII for 4 weeks, renal damage persists and the amount of Cx43 and phosphorylated MYPT1 (p-MYPT) remain elevated. If the treatment with AngII is shorter than 4 weeks, changes in renal function are reversible. When Fasudil, an inhibitor of the RhoA/ROCK pathway is administered 2 weeks before the onset of irreversibility and is maintained for 4 weeks, hypertension persists, but the amount of Cx43 and phosphorylated MYPT-1 return to normal values and renal damage is minimal, demonstrating a relationship between the RhoA/ROCK pathway, the amount of Cx43 and chronic kidney damage.

**Table 1 ijms-20-04408-t001:** Values for weight, proteinuria, creatininuria, creatininemia, creatinine clearance, and fractional excretion (FE) for Na+ and K+ in the experimental groups. Protein and creatinine were measured to assess renal function; the fractional excretion of sodium and potassium (FENa+, FEK+, [%]) was also determined. Data are expressed as mean ± SE. The differences were evaluated by analysis of variance followed by the Tukey post-hoc test. *** *p* < 0.001 vs. AngII groups (*n* ≥ 4/all groups).

Groups	Weigth (gr)	Proteinuria (mg/day)	Creatinine Clearence (ml/min)	FE Na + (%)	FE K + (%)
**Ctrl**	482 ± 31	2.7 ± 1.1 ***	1.4 ± 0.3 ***	0.2 ± 0.0 ***	12.0 ± 2.7 ***
**Ctrll+fasudil**	480 ± 36	3.6 ± 1.1 ***	2.1± 0.1 ***	0.1 ± 0.0 ***	12.5 ± 0.3 ***
**AngII**	364 ± 42	214.0 ± 19.0	0.7 ± 0.0	2.2 ± 0.4	162.0 ± 23.0
**AngII+fasudil**	368 ± 17	19 ± 7.2 ***	1.9 ± 0.2 ***	0.5 ± 0.1 ***	30 ± 7.2 ***

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
