# Peer review of "Role of a RhoA/ROCK-Dependent Pathway on Renal Connexin43 Regulation in the Angiotensin II-Induced Renal Damage"

_ijms, 2019, doi:10.3390/ijms20184408_

Round 1

Reviewer 1 Report

The paper by Gomez et al focuses on Cx43 being regulated by the RhoA/ROCK pathway within an AngII infusion renal damage model in Sprague-Dawley rats which can be saved with fasudil. The phenomenon of angII causing renal damage remains of unclear pathophysiology despite the wide use of ACE inhibitors in clinical practice so the paper is valuable to the scientific community. The paper is in essence ready to be accepted.
I just have two questions:

Major

Do the authors think that a longer treatment with fasudil, for example 4 weeks instead of 2 weeks would result in SBP lowering? Why is fasudil not approved by the FDA and the EMA? Please add a short sentence to the discussion

Author Response

Reviewer 1

Major

Do the authors think that a longer treatment with fasudil, for example 4 weeks instead of 2 weeks would result in SBP lowering?

It is an interesting possibility, but currently we do not know whether longer periods of treatment with Fasudil would improve SBP. This proposed study will be carried out and will be included in a future communication dedicated to elucidating the complete vision of chronic kidney damage and the comment you mention about therapeutic treatment.

Why is fasudil not approved by the FDA and the EMA? Please add a short sentence to the discussion.

Thank you very much for you comment and we have now included a sentence regarding this topic.

Reviewer 2 Report

Gonzalo Gómez et al. continued their work published on IJMS 19, 957 (2018) (note: duplicate references # 31 and #57) on this rat kidney injury model induced by chronic infusion with Ang II along with or without a vasodilator Fasudil, an inhibitor of ROCK, to test whether RhoA/ROCK-dependent pathway in regulation of blood pressure and expression of Cx43. Several parameters such as SBP, macrophage infiltration, OS, inflammation (IL-1b), fibrosis and ROCK activity (p-MYPT1/MYPT1) were also analyzed.

Figs. 1-6 characterized the time (4-weeks) would cause irreversible pathological consequence.

Fig. 7 showed Fasudil has “preventive” effect to alleviate pathological indications.

Fig. 8 is a schematic summary.

The work is interesting, but the quality of manuscript should be much improved before publication.

Major-

It would be interesting to know if Fasudil has “therapeutic” effect as many CKD patients might not be aware of disease state until later stage in CKD progression. Therefore, it is suggested to include data on AngII 4 along with different period of Fasudil treatment to test if Fasudil could revert to normal condition when compared to non-treated animals. In discussion, there is no discussion on why there is no change of SBP after Fasudil treatment. Different possible mechanism should be proposed.  The IHC micrographs only showed small area of tissue. Larger size of micrographs should be shown to better reflect the kidney pathology.

Minor-

P2 third paragraph, full names of AT1 and AT2 should be shown. P3 third paragraph, please make it clear about the context of last (and a very long, which spread 8 lines) sentence. P3 the last paragraph, it should be “starts” but not “stops” recovering…. P4 the first paragraph, the SBP value of Ang 4+2 (150 mmHg) is missing from text. Fig. 1B. It is recommend to use different line color to represent Fasudil treatment (in differentiation from the grey line representing spontaneous recovering). This is also recommended for P13 Fig. 8. Also, please label the black line as Ang treatment. Please clarify Fig. 1B legends as it is hard to understand. P5 Fig. 2B, the decrease of Up/Uc in Ang 5 (31.7) in comparison to Ang 4 (50) and Ang 6 (47) was not explained. P6 the second paragraph, text about Fig. 4B was about plasma IL-1b level, however, the graph shown was IL-1b level from kidney homogenates. Please clarify this. P7 Fig. 4A the right panel of the top row micrographs, it should be labeled as “2+2” but not “4+2”. P7 Fig 4 legends and P10 Fig. 7, please change Il-1 to IL-1b. P8 Fig. 5A the right panel of the top row micrographs, it should be labeled as “2+2” but not “4+2”. P10 Fig. 7, please show scale bars. Again, larger micrographs should be shown instead of small area micrographs. P12 the last paragraph, MES-13 is a mesangial cell line (more specifically). P14 on animal, please describe method of urine collection. P14 on Ang II administration, what is high-dose of Ang II in minipump? P15 on TBARS, please list reference by Ramanahtan et al. P16 on reading of IL-1b level, it should be pg/g protein (not ng/ml). P16 on WB, please indicate antibody sources. P17, ED-1 is marker of macrophage infiltration P17, NOX is NADPH “oxidase” (not NADPH) Duplicate references, ref #31 and #57 are the same.v

Author Response

Reviewer 2

Major

It would be interesting to know if Fasudil has “therapeutic” effect as many CKD patients might not be aware of disease state until later stage in CKD progression. Therefore, it is suggested to include data on AngII 4 along with different period of Fasudil treatment to test if Fasudil could revert to normal condition when compared to non-treated animals.

The comment you suggest I find it very interesting and I thank you for it. This proposed study will be carried out and will be included in a future communication dedicated to elucidating the complete vision of chronic kidney damage and the comment you mention about therapeutic treatment.

In discussion, there is no discussion on why there is no change of SBP after Fasudil treatment. Different possible mechanism should be proposed.

It is difficult to be able to elucidate a clear mechanism that explains the phenomenon shown in this work. In general, the articles that have shown a similar effect to what I have observed do not have a clear explanation either. Anyway, I have added additional evidence that supports probable ways to explain the renoprotective effect observed without having effects in reducing blood pressure.

The IHC micrographs only showed small area of tissue. Larger size of micrographs should be shown to better reflect the kidney pathology.

Taking into account your comment, it was not considered necessary to add a larger size micrograph since it corresponds to the representative photo of 4 different fields and of an n = 3, therefore it is a representative image of a total of 12 micrographs taken. Therefore, including a lower magnification picture would not be adding additional information. Anyway, I made the respective clarification in this new version and of course grateful for your suggestion.

Minor

P2 third paragraph, full names of AT1 and AT2 should be shown.

We agree and it is now included.

P3 third paragraph, please make it clear about the context of last (and a very long, which spread 8 lines) sentence.

It has been clarified and is included in this new version

P3 the last paragraph, it should be “starts” but not “stops” recovering.

It has been clarified and is included in this new version

P4 the first paragraph, the SBP value of Ang 4+2 (150 mmHg) is missing from text.

We agree and it is now included.

1B. It is recommended to use different line color to represent Fasudil treatment (in differentiation from the grey line representing spontaneous recovering).

It has been clarified and is included in this new version

This is also recommended for P13 Fig. 8. Also, please label the black line as Ang treatment. Please clarify Fig. 1B legends as it is hard to understand.

It has been clarified and is included in this new version

P5 Fig. 2B, the decrease of Up/Uc in Ang 5 (31.7) in comparison to Ang 4 (50) and Ang 6 (47) was not explained.

We agree and it is now included.

P6 the second paragraph, text about Fig. 4B was about plasma IL-1b level, however, the graph shown was IL-1b level from kidney homogenates. Please clarify this.

It has been clarified and is included in this new version

P7 Fig. 4A the right panel of the top row micrographs, it should be labeled as “2+2” but not “4+2”.

We agree and it is now included.

P7 Fig 4 legends and P10 Fig. 7, please change Il-1 to IL-1b.

We agree and it is now included.

P8 Fig. 5A the right panel of the top row micrographs, it should be labeled as “2+2” but not “4+2”.

We agree and it is now included.

P10 Fig. 7, please show scale bars. Again, larger micrographs should be shown instead of small area micrographs.

The scale bar is added in included in this new version.

Larger micrographs are were not considered necessary to add a larger size micrograph since it corresponds to the representative photo of 4 different fields and of an n = 3, therefore it is a representative image of a total of 12 micrographs taken.

P12 the last paragraph, MES-13 is a mesangial cell line (more specifically).

It has been clarified and is included in this new version

P14 on animal, please describe method of urine collection.

It has been clarified and is included in this new version

P14 on Ang II administration, what is high dose of Ang II in minipump?

It has been clarified and is included in this new version

P15 on TBARS, please list reference by Ramanahtan et al.

We agree and it is now included.

P16 on reading of IL-1b level, it should be pg/g protein (not ng/ml).

We agree and it is now included.

P16 on WB, please indicate antibody sources.

the antibody specification is in the reagents and antibodies section.

P17, ED-1 is marker of macrophage infiltration

We agree and it is now included.

P17, NOX is NADPH “oxidase” (not NADPH)

We agree and it is now included.

Round 2

Reviewer 2 Report

Fig 1 is very confusing; it now has two sets of A and B diagrams. Please make correction. Fasudil treatment should be represented in red line.

Please make corrections on “IL-1β”.

Author Response

Minor

Fig 1 is very confusing; it now has two sets of A and B diagrams. Please make correction. Fasudil treatment should be represented in red line.

It has been clarified and is included in this new version

Please make corrections on “IL-1β”

It has been clarified and is included in this new version